# Integrated transcriptomic and neuroimaging brain model decodes biological mechanisms in aging and Alzheimer's disease

**Quadri Adewale[1,2,3], Ahmed F Khan[1,2,3], Felix Carbonell[4], Yasser Iturria-Medina[1,2,3]\*, Alzheimer's Disease Neuroimaging Initiative[†]**

[1]Neurology and Neurosurgery Department, Montreal Neurological Institute, McGill University, Montreal, Canada; [2]McConnell Brain Imaging Centre, Montreal Neurological Institute, McGill University, Montreal, Canada; [3]Ludmer Centre for Neuroinformatics and Mental Health, McGill University, Montreal, Canada; [4]Biospective Inc, Montreal, Canada

**\*For correspondence:**
lturria.medina@gmail.com

[†]Data used in preparation of this article were partly obtained from the Alzheimer's Disease Neuroimaging Initiative (ADNI) database (adni.loni.usc.edu). As such, the investigators within the ADNI contributed to the design and implementation of ADNI and/or provided data but did not participate in analysis or writing of this report. A complete listing of ADNI investigators can be found at: http://adni.loni.usc.edu/wpcontent/uploads/how_to_apply/ADNI_Acknowledgement_List.pdf.

**Abstract** Both healthy aging and Alzheimer's disease (AD) are characterized by concurrent alterations in several biological factors. However, generative brain models of aging and AD are limited in incorporating the measures of these biological factors at different spatial resolutions. Here, we propose a personalized bottom-up spatiotemporal brain model that accounts for the direct interplay between hundreds of RNA transcripts and multiple macroscopic neuroimaging modalities (PET, MRI). In normal elderly and AD participants, the model identifies top genes modulating tau and amyloid-β burdens, vascular flow, glucose metabolism, functional activity, and atrophy to drive cognitive decline. The results also revealed that AD and healthy aging share specific biological mechanisms, even though AD is a separate entity with considerably more altered pathways. Overall, this personalized model offers novel insights into the multiscale alterations in the elderly brain, with important implications for identifying effective genetic targets for extending healthy aging and treating AD progression.

## Introduction

Innovations in healthcare and drug delivery have led to increase in human life expectancy. However, increased lifespan is accompanied by more predisposition to frailty and late-onset Alzheimer's disease (AD) (*Guerreiro and Bras, 2015*; *Singh et al., 2019*). Both healthy aging and AD are complex multifactorial processes, and understanding their molecular mechanisms is crucial for extending longevity and improving the quality of life (*Alkadhi and Eriksen, 2011*; *Kowald and Kirkwood, 1996*). Indeed, at the microscopic scale (~$10^{-6}$ m), transcriptomics and proteomics analyses of the brain have paved the way for deciphering the mechanistic underpinnings of healthy aging and AD (*Dillman et al., 2017*; *Iturria-Medina et al., 2020*; *Johnson et al., 2020*; *Mostafavi et al., 2018*; *Tanaka et al., 2018*). In parallel, macroscopic (~$10^{-2}$ m) imaging phenotypes from PET and MRI are facilitating the detailed characterization of brain changes, such as amyloid-β (Aβ) and tau accumulation, glucose hypometabolism, altered cerebrovascular flow, and atrophy (*Dukart et al., 2013*; *Jack et al., 2018*; *Rodrigue et al., 2012*; *Zhang et al., 2017*). However, in both aging and disease research, most studies incorporate brain measurements at either micro- (e.g., transcriptomics) or macroscopic scale (e.g., PET imaging), failing to detect the direct causal relationships between several biological factors at multiple spatial resolutions.

Although AD is characterized by the accumulation of amyloid plaques and neurofibrillary tangles, many other biological aberrations have been associated with the disease (neuroinflammation, vascular abnormalities, white matter hyperintensities), leading to changes in diagnostic criteria in recent times (*DeTure and Dickson, 2019*). The complexity of AD is further compounded by the interplay between these multiple biological factors. A growing body of evidence points to the synergistic interaction between Aβ and tau in driving neuronal loss, functional dysregulation, and glucose hypometabolism in AD (*Iaccarino et al., 2017*; *Ittner and Götz, 2011*; *Pascoal et al., 2017*; *Pickett et al., 2019*). Also, cerebral blood flow (CBF) promotes Aβ clearance, suggesting that vascular dysregulation could impact neuronal function and facilitate Aβ deposition (*Qosa et al., 2014*; *Zlokovic, 2011*). To account for the synergy between multiple biological factors, we previously introduced a multifactorial causal model (MCM) (*Iturria-Medina et al., 2017*), which uses multimodal imaging data to characterize the macroscale intra-regional interactions among any pair of biological factors (e.g., tau, Aβ, CBF) while accounting for the inter-regional spreading of the pathological alterations across axonal and/or vascular connections. However, this multifactorial model did not consider the microscopic properties of the modelled brain regions.

In an initial attempt to integrate brain variables at multiple scales, a few recent studies have used the regional expression patterns of pre-selected genes as complementary information in intra-brain disease-spreading models (*Freeze et al., 2018*; *Freeze et al., 2019*; *Zheng et al., 2019*). Applied to Parkinson's disease (PD), improvements in the capacity to explain regional brain atrophy patterns were observed, based on each brain region's genetic predisposition to the disease. However, most of these studies have selected very specific genes already known for their crucial role in disease (e. g., *SNCA*, *TMEM175*, *GBA*), while disregarding the individual and combined roles of several other relevant gene candidates. Moreover, the analyses have focused on the influence of transcriptomics on a single biological factor at a time, without accounting for the multiplicity of biological alterations and interactions that occur at different spatial scales. As a result, we continue to lack brain generative models integrating a large set of genetic activities with multimodal brain properties.

An integrated multiscale and multifactorial brain model (from genes to neuroimaging and cognition) may be critical to further our understanding of both healthy aging and neurodegeneration, and engender the development of inclusive biomarkers for personalized diagnoses and treatment. Driven by this motivation, here we combine whole-brain transcriptomics, PET, and MRI in a comprehensive generative and personalized formulation, which we successfully validated in healthy aging and AD progression. This novel approach concurrently accounts for the direct influence of hundreds of genes on regional macroscopic multifactorial effects, the pathological spreading of the ensuing aberrations across axonal and vascular networks, and the resultant effects of these alterations on cognition. The proposed framework constitutes a promising technique for identifying effective genetic targets to prevent aging-related disorders and ameliorate existing neurodegenerative conditions.

## Results

### Capturing gene and macroscopic factor interactions in the human brain

Genes control many biological functions, and their dysregulation can cause abnormal development, accelerated aging, or disease (*Kuintzle et al., 2017*; *Lee and Young, 2013*). Aiming to characterize the direct influence of genes on multiple brain processes, here we have developed a multiscale and multifactorial spatiotemporal brain model (*Figure 1A–C*) linking whole-brain gene expression with multiple macroscopic factors typically quantified via molecular PET and MRI modalities (i.e., Aβ and tau proteins, CBF, glucose metabolism, neuronal activity, and grey matter density). This novel approach, called Gene Expression Multifactorial Causal Model (GE-MCM; see 'Methods'), enables the quantification of gene-specific impacts on the longitudinal changes associated with each local macroscopic factor considered and gene-mediation effects on pairwise factor interactions (e.g., negative tau effects on neuronal activity) while accounting for the simultaneous spreading of the aberrant effects across physical brain networks (e.g., tau and Aβ region-region propagation via anatomical and vascular connectomes). By using standardized gene expression (GE) maps (*Hawrylycz et al., 2012*), longitudinal multimodal imaging data, and a robust optimization algorithm, the GE-MCM identifies individual transcriptomic-imaging parameters controlling the dynamic

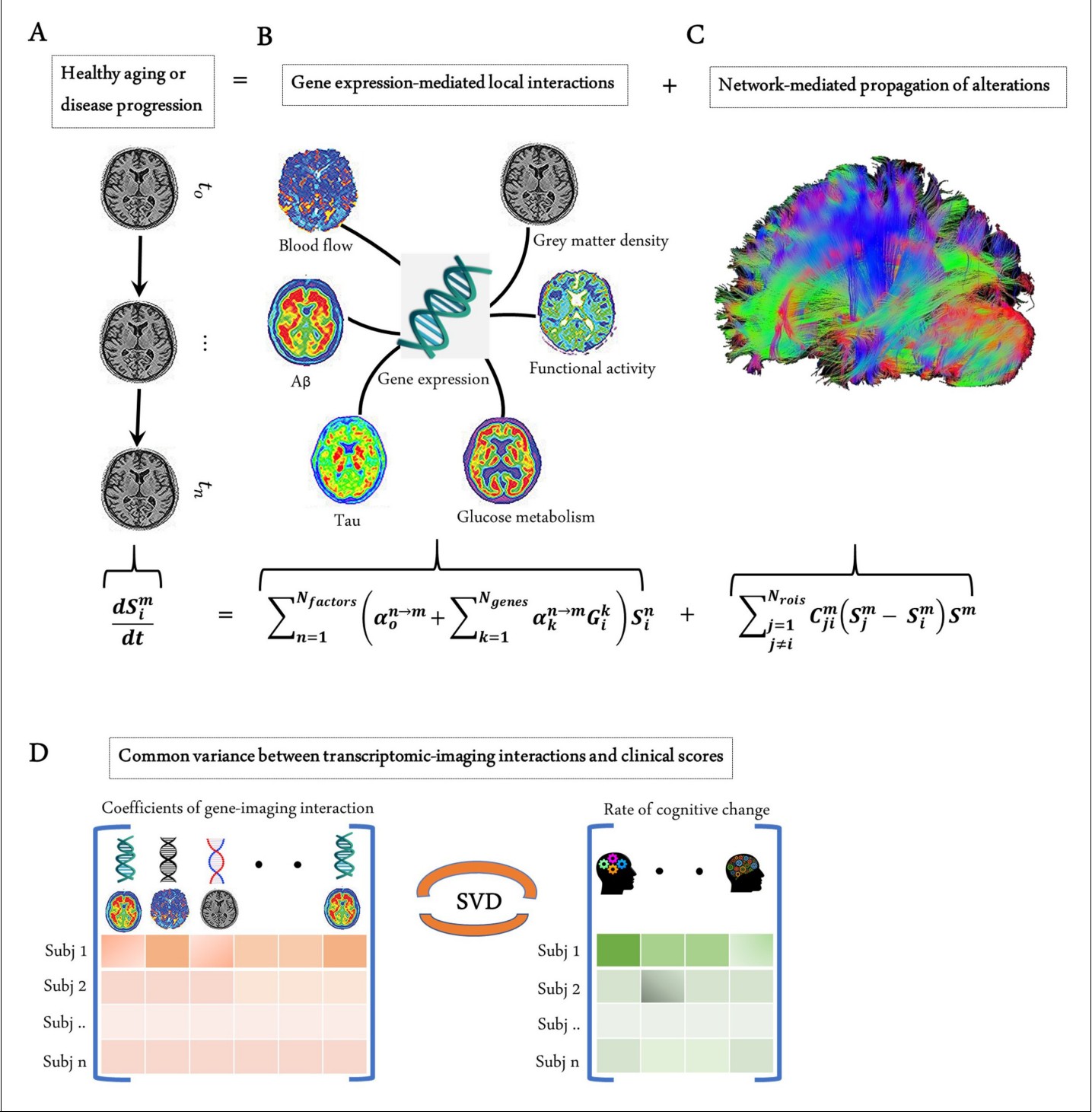

**Figure 1.** Modelling the gene-imaging interactions driving healthy aging and AD progression. (**A**) The longitudinal alteration of macroscopic biological factors in healthy and diseased brain due to gene-imaging interactions and the propagation of the ensuing alterations across brain network. (**B**) Regional multifactorial interactions between six macroscopic biological factors/imaging modalities are modulated by local gene expression. (**C**) Causal multifactorial propagation network capturing the interregional spread of biological factor alterations through physical connections. (**D**) By applying a multivariate analysis through singular value decomposition (SVD), the maximum cross-correlation between age-related changes in cognitive/clinical evaluation and the magnitude of genetic modulation of imaging modalities are determined in a cohort of stable healthy subjects (for healthy aging), mild cognitive impairment (MCI) converters, and Alzheimer's disease (AD) subjects (for AD progression). The key causal genes driving healthy aging and AD progression are identified through their absolute contributions to the explained common variance between the gene-imaging interactions and cognitive scores.

*Figure 1 continued on next page*

**Figure supplement 1.** Subject selection.
**Figure supplement 2.** Correlation of predicted mRNA expression with actual mRNA expression across 976 genes.
**Figure supplement 2—source data 1.** Source data for *Figure 1—figure supplement 2*.

changes observed in the macroscopic biological factors considered (*Figure 1A–C*). These personalized parameters are assumed to be the gene-specific deviations required for model fitting and, thus, they quantitatively measure individual gene dysregulation patterns. We hypothesized that the post-hoc analysis of these transcriptomic-imaging parameters will reveal essential pathogenetic mechanisms in health and disease.

Next, with the complementary interest of further clarifying the genetic mechanisms underlying healthy aging and AD development, the GE-MCM framework was applied to a cohort of 151 healthy and 309 diseased subjects from Alzheimer's Disease Neuroimaging Initiative (ADNI) (see 'Methods' and *Figure 1*). The standardized transcriptomic data was derived from six neurotypical brains from Allen Human Brain Atlas (AHBA) (*Hawrylycz et al., 2012*), comprising RNA intensities of 976 landmark genes with leading roles in central biological functions. These genes correspond to a set of universally informative transcripts, previously identified as 'Landmark Genes', based on their capacity to cover most of the information in the whole human transcriptome across a diversity of tissue types (*Subramanian et al., 2017*).

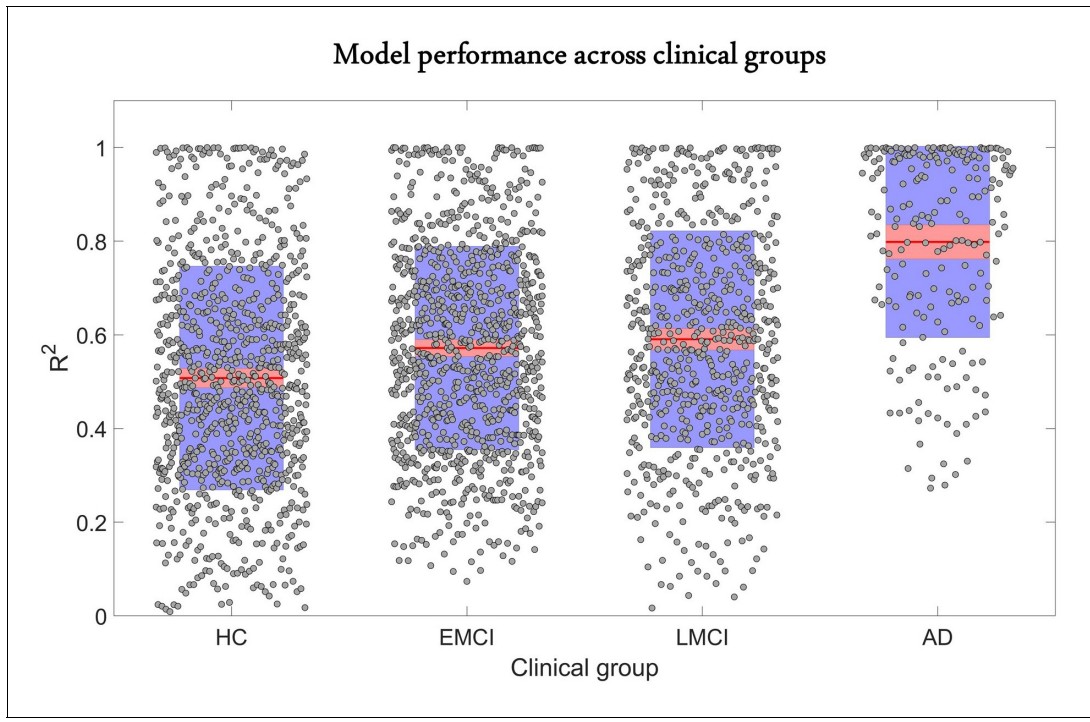

**Figure 2.** Reconstruction of individual multifactorial alteration patterns across all subjects in the AD continuum. Plots are shown for the $R^2$ obtained across all six biological factors in the healthy control (HC) (n=151), early mild cognitive impairment (EMCI) (n=161), late mild cognitive impairment (LMCI) (n=113), and Alzheimer's disease (AD) (n=35) cohorts. Points are laid over a 2.58 standard error of the mean (SEM) (99% confidence interval) in red and at 1 SD in blue. Notice that model performance improves with disease progression. We attribute this effect to the typical larger variation in longitudinal biological factor alterations with disease evolution, which provides the optimization algorithm with further biological information and results in a more accurate data fitting and parameter identification.

The online version of this article includes the following source data for figure 2:

**Source data 1.** Source data for *Figure 2*.

The predictive performance of the model across different clinical categories is shown in *Figure 2*. We calculated the coefficient of determination ($R^2$) of the model for the six longitudinal PET and MRI modalities, and averaged them across all subjects in each clinical group. The $R^2$ was highest for AD ($0.80 \pm 0.20$), followed in order by late mild cognitive impairment (LMCI) ($0.59 \pm 0.23$), early mild cognitive impairment (EMCI) ($0.57 \pm 0.21$), and healthy control (HC) ($0.51 \pm 0.24$). The improvement observed in model performance with disease progression could be due to the larger variation in biological factor alteration in the later stages of the AD continuum. Nevertheless, these results support the capacity of the GE-MCM approach to reproduce the longitudinal observations in the six molecular PET and MRI modalities.

## Identifying genes driving biological and cognitive changes in healthy aging

Age is a significant risk factor for developing many complex disorders. Even though lifestyle and environmental factors contribute to healthy aging, understanding the genetic basis of aging will offer valuable biological insights with implications for disease prevention and longevity (*Niccoli and Partridge, 2012*; *Rodríguez-Rodero et al., 2011*). Hence, we sought to identify causal genes underlying longitudinal cognitive changes in healthy aging. We analysed the predictive relationship between the obtained transcriptomic-imaging parameters and multiple cognitive evaluations in 113 HC subjects who remained clinically stable within 7.8 years (SD = 2.9 years). The cognitive changes were obtained as the age-related slopes of Mini-Mental State Examination (MMSE), Alzheimer's Disease Assessment Scale (ADAS), executive function (EF), and memory composite score (MEM) over an average of 7.2 time points (SD = 2.6). For this analysis, we only used 68 stable transcriptomic-imaging parameters, the 99% CI of which excluded zero across the HC non-converters ('Model evaluation' subsection in 'Methods'). Using a multivariate singular value decomposition (SVD), we found the common latent variables between the gene-imaging parameters and the slopes of multiple cognitive measures, and the variances explained by the principal components (PCs) are shown in *Figure 3A*. Running 10,000 permutations, we identified the first PC as the only significant component (explained variance = 50.3%; p=0.0074).

Next, we calculated the contribution of each gene-specific parameter on this significant PC ('Model evaluation') and assessed the statistical reliability of the genetic contributions by running 10,000 bootstrap iterations. A bootstrap ratio threshold of 2.58 (approximately equivalent to p<0.01; *Efron and Tibshirani, 1986*) was applied, revealing eight genes with stable causal contributions to the multimodal imaging dynamics and associated cognitive changes in healthy aging (*Figure 3B*). Notice that the saliences of some genes are negative, implying that their modulation effects are negatively associated with the rate of cognitive change. Specifically, as shown in *Figure 3C*, *TSKU* modulates Aβ while tau is modulated by *GNA15* and *LSM6* to drive age-related alterations in Aβ. Also, *BIRC5*, *SESN1*, and *PLSCR3*, respectively, modulate tau, CBF, and Aβ in driving alterations in neuronal activity. Similarly, age-related changes in tau are driven by *C5* and *CASP10* through their direct effects on functional activity and CBF, respectively.

## Revealing top genes and molecular pathways controlling multifactorial alterations and clinical deterioration in AD

A crucial challenge for the early detection and prevention of AD is the development of cheap and non-invasive biomarkers (such as genes) as well as the understanding of the molecular mechanisms underlying its pathogenesis (*Iturria-Medina et al., 2020*). Here, we proceed to identify genes driving neuropathological progression in the AD spectrum, restricting our analysis to 129 participants who were either diagnosed with AD (35) at baseline or converted to AD (94) after baseline diagnosis (7 HC and 87 MCI). Like the aging analysis, we only kept 993 statistically stable transcriptomic-imaging parameters, the 99% CI of which excluded zero ('Model evaluation' subsection in 'Methods'). We used SVD to obtain the common latent variables (variance) between the gene-imaging parameters and slopes of multiple cognitive measures (MMSE, ADAS, EF, and MEM across $6.3 \pm 3.0$ longitudinal time points). After 10,000 permutation runs, the first PC was significant (p=0.009) and explained 63.8% of the variance between the gene-imaging interaction parameters and the slopes of cognitive evaluations (*Figure 4A*). A bootstrap ratio threshold of 2.58 (approximately equivalent to p<0.01; *Efron and Tibshirani, 1986*) was applied, identifying 111 genes (*Figure 4B*) with stable causal

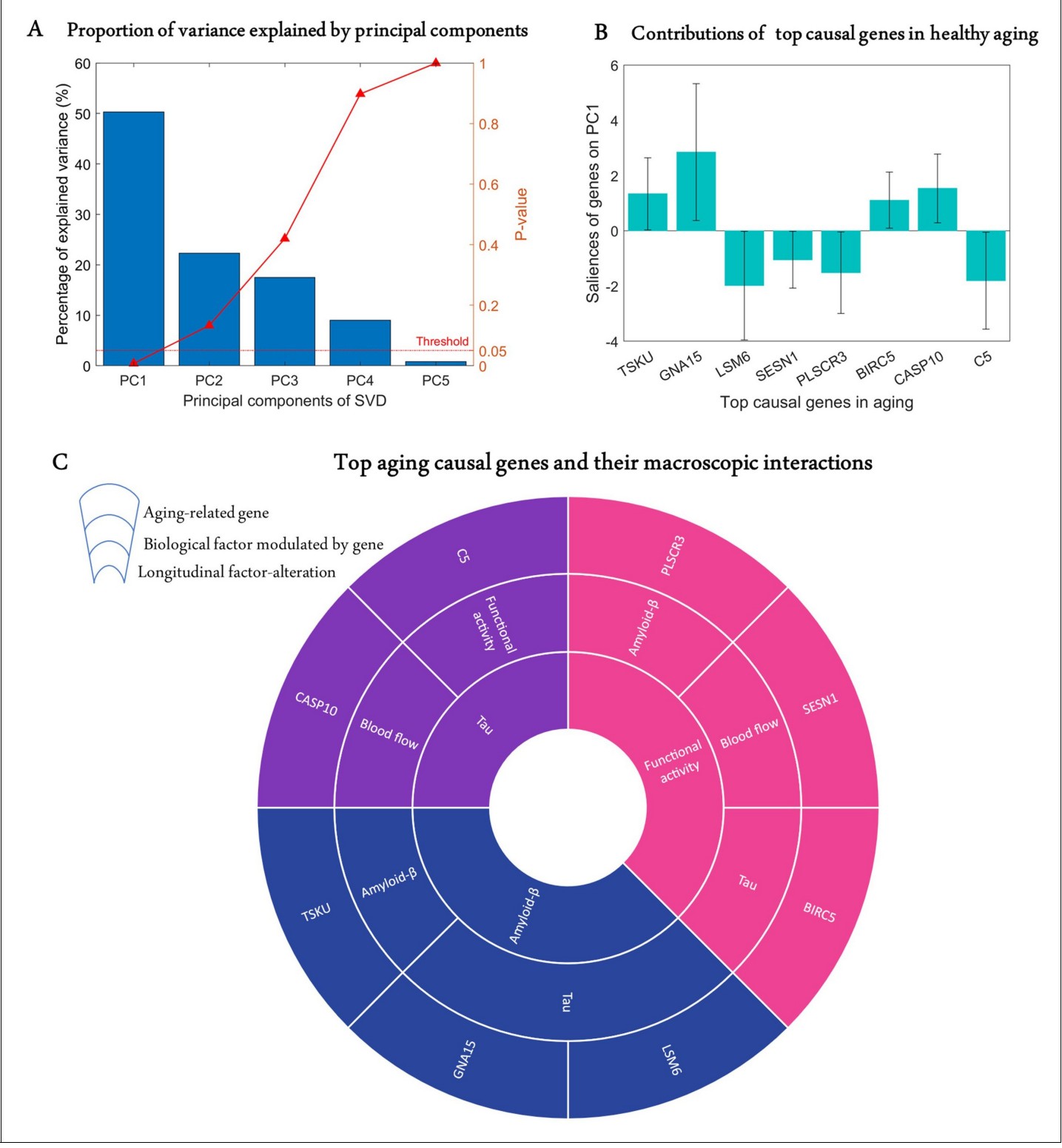

**Figure 3.** Identification of top genetic modulators of cognitive change in healthy aging. (**A**) Common variance (and associated p-values) captured by the top five principal components (PCs) of the singular value decomposition (SVD) in explaining the rate of change of cognitive scores due to healthy aging. Only the first PC is significant (p<0.05). (**B**) Genetic contributions (and 99% CI) on the first PC, depicted only for the eight highly stable aging-related genes, the bootstrap ratios of which are above 2.58. (**C**) Top genetic determinants of multifactorial alterations in healthy aging. The innermost ring shows the longitudinal biological factor altered with aging, the middle ring displays the interacting biological factors driving the longitudinal
*Figure 3 continued on next page*

*Figure 3 continued*
alteration, and the outermost ring represents the causal genes modulating the interactions among biological factors (e.g., *SESN1* directly modulates blood flow to drive age-related alteration in neuronal activity).

The online version of this article includes the following source data for figure 3:

**Source data 1.** Source data for *Figure 3*.

contributions to the macroscopic factor interactions and associated cognitive changes in AD. The factors directly modulated by these causal genes and the ensuing factorial alterations are shown in *Figure 4C*.

Finally, we performed a large-scale gene functional analysis with PANTHER (*Mi et al., 2013*) to uncover the molecular pathways and biological functions associated with the 111 identified disease-driving genes. Sixty-five functional pathways were identified and most of them, including the Alzheimer disease-presenilin pathway, are highly representative of the biological processes commonly associated with neuropathology and cognitive decline (*Supplementary file 5*). The pathways with the leading number of genes are apoptosis, cholecystokinin receptor signalling, inflammation mediated by chemokine and cytokine, and gonadotropin-releasing hormone receptor (see 'Discussion').

## Discussion

### Gene expression patterns modulate multifactorial interactions in healthy aging and AD progression

An unprecedented attribute of this study is the insight it provides into the multiscale interactions among aging and AD-associated biological factors, and the possible mechanistic roles of the identified genetic determinants. In concordance with our results in healthy aging (*Figure 3C*), *BIRC5* has been shown to regulate microtubule dynamics and interact with tau (*Zhao et al., 2003*). Sestrins, including *SESN1,* preserve blood brain barrier integrity and serve a neuroprotective effect after cerebral ischaemia (*Chen et al., 2019*; *Li et al., 2016*; *Shi et al., 2017*). *C5* belongs to the complement immune system, and it modulates synaptic pruning and plasticity by interacting with microglia (*Wang et al., 2020*).

Several animal and biostatistical studies also corroborate the functional relationships observed in AD results. In agreement with the interactions driving longitudinal alterations in blood flow (*Figure 4C*), *FKBP4* encodes the *FKBP52* protein, which has been demonstrated to alter tau phosphorylation pattern and stimulate its abnormal aggregation (*Giustiniani et al., 2015*). *FKBP52* also decreased significantly in brains of AD patients (*Giustiniani et al., 2012*). A bioinformatic and functional validation study identified the role of *GNAS* in glucose metabolism through insulin regulation (*Taneera et al., 2019*). Notably, several studies have consistently linked *MEF2C* to AD and its associated cognitive decline (*Beecham et al., 2014* ; *Davies et al., 2015*; *Sao et al., 2018*). Knocking out *MEF2C* in mice induced glucose metabolism impairment (*Anderson et al., 2015*). *PLSCR1* could drive atrophy due to its apoptotic effect and interaction with calcium ion in maintaining the organization of phospholipid bilayers of membranes (*Sahu et al., 2007*). *CXCR4* also regulates apoptosis and neuronal survival through glial signalling and the Rb/E2F pathway, respectively (*Bezzi et al., 2001*; *Khan et al., 2008*). Nitric oxide synthase interacting protein (*NOSIP*) controls the expression of nitric oxide synthase (NOS), the major source of nitric oxide in the brain (*Dreyer et al., 2004*). In brain endothelial cells, downregulating NOS upregulates *APP* (amyloid precursor protein) and *BACE1* (β-site APP-cleaving enzyme1), both of which control amyloid dynamics (*Austin et al., 2010*).

We also found congruous functional associations for the genes driving longitudinal alterations in Aβ. Apart from its apoptotic role, *CASP3* has been shown to regulate synaptic plasticity and functional activity in vivo (*D'Amelio et al., 2010*). *TRIB3* controls glucose metabolism, insulin signalling, and the expression of other glucose metabolism genes (*Prudente et al., 2012*; *Zhang et al., 2013*; *Zhang et al., 2016*). Among the genes altering tau with AD progression, nuclear factor of activated T cells (*NFAT*) overexpression in animal model increased Aβ production and promoted *BACE1* transcription (*Mei et al., 2015*). TIMELESS (*TIM*) is a gene with central role in controlling circadian neuronal activity (*Kurien et al., 2019*). Interestingly, dysregulated circadian rhythm is causally associated

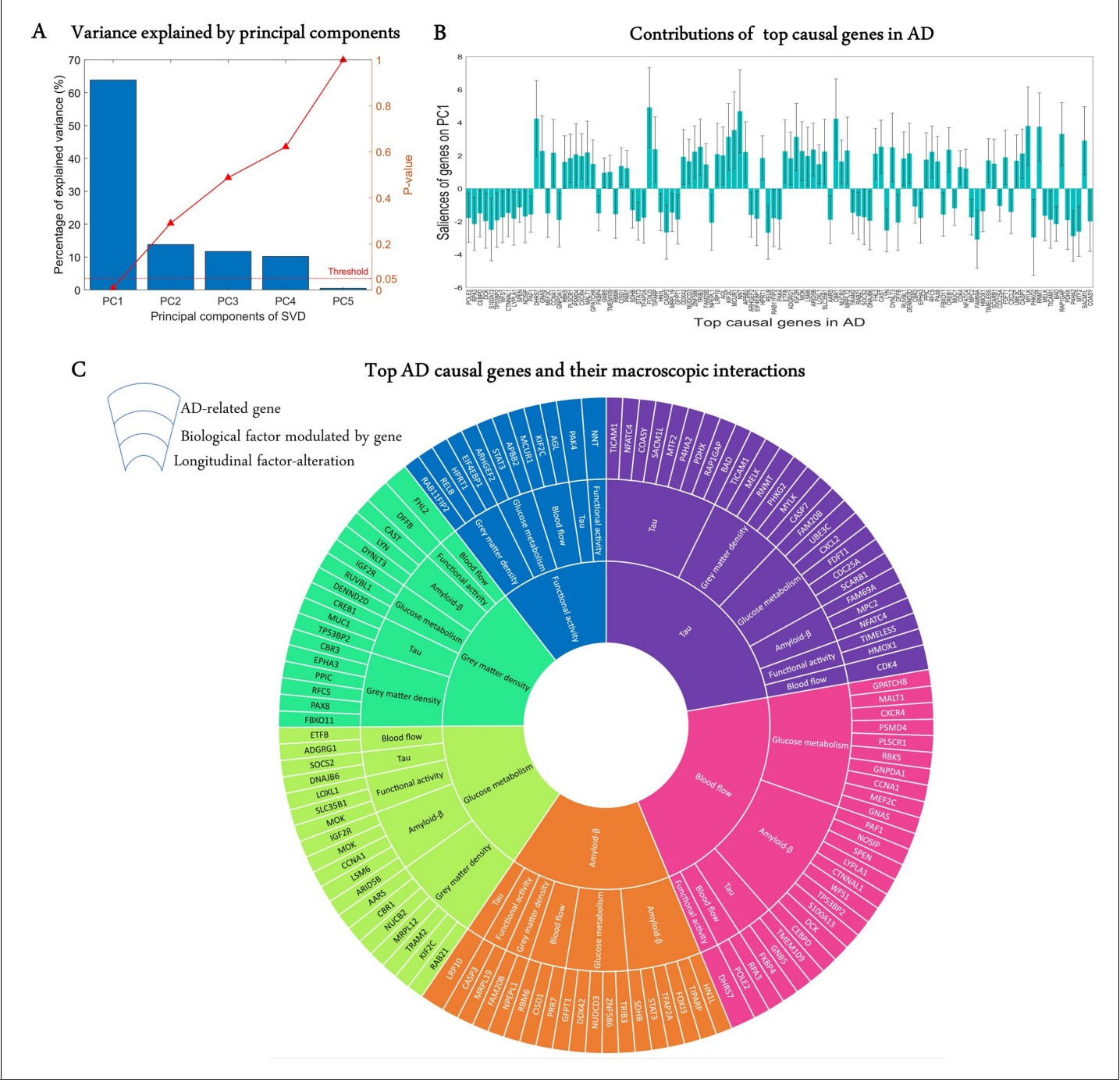

**Figure 4.** Uncovering the top genetic determinants of AD progression. (**A**) The common variance captured by the principal components (PCs) of the singular value decomposition (SVD) in explaining how clinical evaluations change with Alzheimer's disease (AD) evolution. P-values after 10,000 permutations are also shown. (**B**) Contributions of top AD causal genes (with 99% CI) to the first PC. Top causal genes are identified by selecting those genes whose bootstrap ratios of saliences are above 2.58. (**C**) Multifactorial interactions between the identified genes and imaging modalities. The innermost ring shows the longitudinal biological factor changes with AD, the middle ring displays the interacting biological factors driving the longitudinal alteration, and the outermost ring represents the causal genes modulating the interactions among biological factors. A gene directly influences how a biological factor interacts with other factors to cause a factorial alteration along the disease's course.

The online version of this article includes the following source data for figure 4:

**Source data 1.** Source data for *Figure 4*.

with AD (*Homolak et al., 2018*). Furthermore, our results on glucose metabolism dysregulation align with previous functional studies. *RAB21* may induce atrophy through apoptosis and cell growth inhibition (*Ge et al., 2017*). Due to its function in detoxifying reactive aldehydes produced from lipid peroxidation, the carbonyl reductase enzyme *CBR1* could prevent oxidative stress-induced atrophy (*Maser, 2006*). *DNAJ* proteins belong to the group of chaperones that regulate protein homeostasis, and an earlier study has implicated *DNAJB6* in α-synuclein aggregation (*Aprile et al., 2017*). Investigating the effect of *DNAJB6* on tau processing as suggested by our result could provide further insight into the roles of the gene in AD.

Supporting our results for longitudinal alterations in functional activity, downregulating *EIF4EBP1* prevents toxin-induced neuronal atrophy in PD models by blocking the action of apoptotic caspsase-3 (*Xu et al., 2014*). The gene also mediates synaptic reorganization and refinement, independent of post synaptic activity (*Chong et al., 2018*). Even though *APBB2* (amyloid beta A4 precursor protein-binding, family B, member 2) primarily binds to *APP*, knocking out *APBB2* in mice causes glucose intolerance and β-cell dysfunction (*Ye et al., 2018*). In transgenic mice, deleting *STAT3* in β cells and neurons impaired glucose metabolism (*Cui et al., 2004*). *STAT3* also regulates liver glucose homeostasis by modulating the expression of gluconeogenic genes (*Inoue et al., 2004*). A gene co-regulatory network analysis identified *RAB11FIP2* as a differentially expressed gene in axon regeneration, suggesting its possible role in atrophy (*Su et al., 2018*). Correspondingly, a growing body of evidence supports the gene-imaging interactions we found in longitudinal alterations in atrophy. *CAST* overexpression was shown to reduce amyloid burden due to its effect on *BACE1* processing of A*PP* (*Liang et al., 2010*; *Morales-Corraliza et al., 2012*). *FHL2* prevents inflammatory angiogenesis and regulates the function of vascular smooth muscle cells, suggesting its role in blood flow (*Chen et al., 2020*; *Chu et al., 2008*). *IGF2R* (insulin-like growth factor two receptor) interacts with insulin receptors for energy homeostasis, and the dysregulation of the gene is associated with type 2 diabetes (*Chanprasertyothin et al., 2015*). *RUVBL1* is an ATPase that modulates insulin signalling, and *RUVBL1* knock-out mice displayed impaired glucose metabolism (*Mello et al., 2020*).

## Aging and Alzheimer's disease have both common and distinct mechanisms

In this study, we used a single gene expression template for all the subjects due to the unavailability of individual whole-brain gene expression. However, it has to be noted that even though this template has spatial but no temporal variation, for each gene, a model parameter controls its interaction (at the individual level) with each time-varying neuroimaging modality (i.e., the estimated transcriptomic-imaging parameters). At the individual level, the fitted gene-imaging parameters are assumed to reflect the gene-specific deformations required to fit the data. Consequently, these parameters represent quantitative measures of the individual dysregulation or deviation in gene expression patterns and, when analysed across the entire population (e.g., via SVD analysis), the parameters can be used to detect cognitive/clinical related genetic associations. Thus, under normal aging, the parameters obtained from the model optimization should be close to zero. Interestingly, it was observed that only ~70 parameters (out of over 35,000 gene-imaging interaction parameters) were significantly different from zero across the healthy aging population. Conversely, ~1000 parameters significantly differed from zero across the diseased population. We attribute the greater number of significant parameters in AD to more genetic dysregulations and biological mechanism alterations in the disorder (*Iturria-Medina et al., 2020*; *Mostafavi et al., 2018*).

The mechanisms of healthy aging and AD substantially overlap even though AD-related alterations are often accelerated, and the regions of alteration could be different (*Toepper, 2017*; *Xia et al., 2018*). Among the aging-associated genes, *CASP10*, *BIRC5*, and *PLSCR3* are involved in caspase-dependent apoptosis. Interestingly, apoptotic genes were also found in AD, including *CASP3*, *CASP7*, *PLSCR1*, *CREB1*, *RELB*, *IGF2R,* and *DFFB*. Sestrin (*SESN1)* is implicated in oxidative signalling, aging inhibition, and exercise mediation (*Budanov et al., 2010*; *Kim et al., 2020*; *Yang et al., 2013*). Correspondingly, some AD causal genes, including *MEF2C*, *CBR1*, and *NOSIP,* are known for their roles in oxidative stress, supporting the relevance of this pathway to both normal and pathological aging (*Kim et al., 2014*; *Rochette et al., 2013*). Given that G protein-coupled receptors (GPCRs) mediate the cellular response to most hormones/neurotransmitters (*de Oliveira et al., 2019*; *Thathiah and De Strooper, 2011*), it is unsurprising that GPCR-related genes converge on normal aging (*GNA15)* and Alzheimer's disease (*GNAS*, *GNB5*). Detection of some inflammation-

associated genes in AD and the complement component *C5* in aging suggests that immune/inflammatory response change is part of both healthy aging and AD. Indeed, apart from the overlapping pathways, *LSM6* was the only gene common to both normal aging and AD. *LSM6* regulates gene expression and mRNA splicing, and a proteomic study linked its expression level to aging in human muscle cells (*Ubaida-Mohien et al., 2019*). Although altered mRNA splicing is associated with AD (*Johnson et al., 2018*; *Koch, 2018*; *Twine et al., 2011*), a functional validation can further reveal the exact role of *LSM6* in the disease.

## Towards a genetic approach to extending healthy aging and treating Alzheimer's disease

The complexity of aging and the mixed aetiology of neurodegeneration necessitate an integrative multifactorial paradigm. In this study, we advanced the understanding of aging and AD pathology through the mechanistic modelling of how gene activity modulates relevant biological factors (e.g., tau, Aβ, CBF, neuronal activity) to drive the cognitive alterations typically observed in the associated populations. The obtained results, in line with relevant molecular and imaging literature, highlight the strength of our approach by confirming previously identified aging- and AD-associated genes and uncovering new genes with relevant pathophysiological roles. In essence, this flexible formulation directly decodes the genetic mediators of spatiotemporal macroscopic brain alterations with aging and disease progression. Consequently, this work has important implications for the mechanistic understanding of aging and AD pathogenesis and, importantly, for the implementation of a biologically defined patient stratification for personalized medical care.

Current approaches to AD treatment do not account for patient heterogeneity, and such non-personalized methods may not only be ineffective but also can cause undesired secondary effects in patients (*Iturria-Medina et al., 2018*). In a previous study, we used a similar imaging-based framework to show that some patients may need interventions targeting either tau, Aβ, CBF, or metabolism, while others can require a combinatorial therapy (e.g., concurrently targeting tau, Aβ, and metabolic dysregulation) (*Iturria-Medina et al., 2018*). Based on this extended approach (GE-MCM), a gene therapy could replace the single and combinatorial treatment fingerprints described, by targeting highly influential genes modulating those factors in individuals. Many of the gene-imaging relationships found in our study have been previously reported in vivo, and the novel associations can be validated through experimental models. Understanding these relationships is crucial for effective drug development and administration. For instance, we found that *APBB2* is a mediator of glucose metabolism. Thus, metabolic side effects may be considered when selecting *APBB2* as a therapeutic target of amyloid processing.

We have used inferred mRNA values for unobserved regions due to the unavailability of high-spatial-resolution GE data. Nevertheless, the correlations between observed and predicted mRNA values are very high for majority of the genes (*Figure 1—figure supplement 2*), further supporting the feasibility of interpolating mRNA values based on spatial dependence (*Gryglewski et al., 2018*). It is, however, noteworthy that some genes with low correlation values might have low spatial dependence or error in the assay. There is an inherent bias in the merged gene expression data from AHBA due to individual variability, and the AHBA subjects are not very representative of the typical age range in the ADNI cohort. Nevertheless, animal and human studies have reported large concordance between peripheral and brain gene expression, implying that blood gene expression may be used as a surrogate for gene expression in brain tissue (*Iturria-Medina et al., 2020*; *Jasinska et al., 2009*; *Sullivan et al., 2006*; *Witt et al., 2013*). Thus, our future work will therefore focus on using personalized gene expression data from blood samples. The applicability and generalizability of the current formulation would also be tested in other neurological conditions (e.g., Parkinson's disease and frontotemporal dementia).

## Materials and methods

### Data description and processing

#### Study participants

This study involved 944 individuals with six multimodal brain imaging from ADNI (RRID:SCR_003007) (http://adni.loni.usc.edu/; *Figure 1—figure supplement 1*). First, for each imaging modality, a

multivariate outlier identification was performed based on the Mahalanobis distance, with a significant squared distance (p<0.05) denoting an outlier (*Iturria-Medina et al., 2016*). From the 911 subjects that survived outlier detection, we chose 509 subjects having at least four imaging modalities (between amyloid PET, tau PET, glucose metabolism PET, resting-state fMRI, cerebral blood flow ASL, and structural MRI). Then, 460 subjects with at least three time points in any of the imaging modalities were selected for our analyses. Next, for each of these subjects (N = 460), missing imaging modalities at each time point having actual individual data were automatically imputed using the trimmed scores regression with internal PCA (*Folch-Fortuny et al., 2016*). The accuracy of the imputation was validated with a leave-one-out cross-validation (e.g., tau imaging data can be significantly recovered for each subject with actual data, $p<10^{-6}$). Hence, all the 460 subjects used in subsequent analyses have completed all six neuroimaging modalities and an average of 4.7 (±2.5) longitudinal time points. Please see *Figure 1—figure supplement 1* for a detailed flowchart of subject selection and *Supplementary file 1* for demographic characteristics. Among the 460 participants, 151 were clinically identified as asymptomatic or HC, 161 with EMCI, 113 with LMCI, and 35 with probable AD.

## Whole-brain gene expression data and brain parcellation

Microarray data was downloaded from the AHBA (RRID:SCR_007416) website (http://www.brain-map.org) (*Hawrylycz et al., 2012*). The AHBA data consists of mRNA expression in 3702 tissue samples obtained from six neurotypical adult brains. The data were preprocessed by the Allen Institute to reduce the effects of bias due to batch effects. Description of the processing steps can be found in the technical white paper (*Allen Human Brain Atlas, 2013*). For each brain, there are 58,692 probes representing 20,267 unique genes. Transcriptome shows spatial dependence, with adjacent regions having similar expression pattern values (*Gryglewski et al., 2018*). Gaussian kernel regression affords a method of predicting gene expression values for unobserved regions based on the mRNA values of proximal regions. The regression is done as a weighted linear combination of unobserved mRNA, with the weight decreasing outward from proximal to distal regions. In order to select a representative probe for genes with multiple probes, Gaussian kernel regression was applied to predict the mRNA intensity in each of the 3702 samples in MNI space (*Evans et al., 1994*) using leave-one-out cross-validation. The probe with the highest prediction accuracy (among the multiple probes for a gene) was chosen as the representative probe for that gene. Next, because GE values were not available for all the grey matter voxels of the brain, Gaussian kernel regression was also used to predict the GE for the remaining MNI coordinates without mRNA expression intensity. Thus, the whole-brain GE data was obtained for the selected 20,267 probes/genes. It was infeasible to use these ~20,000 AHBA genes for modelling, we therefore selected 976 AHBA genes that can be found in the list of 978 landmark genes identified by *Subramanian et al., 2017*. These landmark genes are universally informative transcripts with the capacity to cover most of the information in the whole human transcriptome across a diversity of tissue types (*Supplementary file 2*).

The brain was parcellated into 144 grey matter regions, and the average expression value of each gene was calculated for each region. The brain parcellation was derived from a combination of two atlases: 88 regions were identified through cytoarchitecture from Julich atlas (*Palomero-Gallagher and Zilles, 2019*) and 56 regions were derived from Brodmann atlas. Six regions were excluded due to zero or strong outlier PET imaging signals in their volumes. The remaining 138 regions were used for analyses (*Supplementary file 3*).

## Cognitive and clinical evaluations

The participants were characterized cognitively using MMSE, MEM, EF (*Gibbons et al., 2012*), and ADAS-Cognitive Subscales 11 and 13 (ADAS-11 and ADAS-13, respectively). They were also clinically diagnosed at baseline as HC, EMCI, LMCI, or probable AD patient.

## Multimodal imaging modalities

### ASL MRI

Resting arterial spin labeling (ASL) data were acquired using the Siemens product PICORE sequence (N = 213) with acquisition parameters TR/TE = 3400/12 ms, TI1/TI2 = 700/1900 ms, FOV = 256 mm, 24 sequential 4-mm-thick slices with a 25% gap between the adjacent slices, partial Fourier factor = 6/

8, bandwidth = 2368 Hz/pix, and imaging matrix size = 64 × 64. The data were processed in six steps as follows: (1) motion correction, (2) perfusion-weighted images (PWI) computation, (3) intensity scaling, (4) CBF image calculation, (5) spatial normalization to MNI space (*Evans et al., 1994*) using the registration parameters obtained for the structural T1 image with the nearest acquisition date, and (6) the mean CBF calculation for each of the considered brain regions. Details of the processing can be found at http://www.adni.loni.usc.edu under 'UCSF ASL Perfusion Processing Methods'.

## Amyloid-$\beta$ PET
A 370-MBq bolus injection of AV-45 was administered to each subject and, after about 50 min, 20-min continuous brain PET imaging scans were acquired (N = 459). The images were reconstructed immediately after the scan and, when motion artifact was detected, another 20-min continuous scan was acquired. The acquired PET scans were then preprocessed using the following four main steps as described in *Jagust et al., 2010*: (1) dynamic co-registration to reduce motion artifacts, (2) across-time averaging, (3) re-sampling and reorientation of scans from native space to a standard voxel image grid space ('AC-PC' space), and (4) spatial filtering to convert the images to a uniform isotropic resolution of 8 mm FWHM. Finally, using the registration parameters obtained for the structural T1 image with the nearest acquisition date, all Aβ scans were transformed to the MNI space (*Evans et al., 1994*). Using the cerebellum as an Aβ non-specific binding reference, standardized uptake value ratio (SUVR) values were calculated for the 138 brain regions under consideration.

## Resting-state fMRI
Resting-state fMRI scans were acquired using an echo-planar pulse sequence on a 3.0T Philips MRI scanner (N = 148) with the following parameters: 140 time points, repetition time (TR) = 3000 ms, echo time (TE) = 30 ms, flip angle = 80°, number of slices = 48, slice thickness = 3.3 mm, spatial resolution = 3×3×3 mm$^3$, and in-plane matrix size = 64 × 64. The scans were corrected for motion and slice timing. Then, they were spatially normalized to MNI space (*Evans et al., 1994*) using the registration parameters obtained for the structural T1 image with the nearest acquisition date. Signal filtering was performed to retain only low-frequency fluctuations (0.01–0.08 Hz) (*Chao-Gan and Yu-Feng, 2010*). Fractional amplitude of low-frequency fluctuation (fALFF) was calculated as a regional quantitative indicator of the brain's functional integrity. fALFF has been shown to be highly sensitive to disease progression (*Iturria-Medina et al., 2016*).

## Fluorodeoxyglucose PET
A 185-MBq (5 + 0.5 mCi) bolus injection of [18F]-FDG was administered to each subject and brain PET imaging data were obtained approximately 20 min after injection (N = 455). The images were attenuation-corrected and then preprocessed as follows *Jagust et al., 2010*: (1) dynamic co-registration of frames to reduce the effects of patient motion, (2) across-time averaging, (3) reorientation from native space to a standard voxel image grid ('AC-PC'), and (4) spatial filtering to convert the images to a uniform isotropic resolution of 8 mm FWHM. Next, using the registration parameters obtained for the structural T1-weighted image with nearest acquisition date, the FDG-PET images were normalized to the MNI space (*Evans et al., 1994*). The cerebellum was then used as a reference to calculate SUVR values for the 138 regions (*Klein and Tourville, 2012*).

## Structural MRI
Structural T1-weighted 3D images were obtained for all subjects (N = 460) as described in http://adni.loni.usc.edu/methods/documents/mri-protocols/. The images were corrected for intensity non-uniformity using the N3 algorithm (*Sled et al., 1998*). Next, they were segmented into grey matter (GM), white matter (WM), and cerebrospinal fluid (CSF) probabilistic maps, using SPM12 (http://www.fil.ion.ucl.ac.uk/spm). The grey matter segmentations were transformed into the MNI space (*Evans et al., 1994*) using DARTEL (*Ashburner, 2007*). To preserve the initial amount of tissue volume, each map was corrected for the effects of the spatial registration. Mean grey matter density and determinant of the Jacobian (DJ) (*Ashburner, 2007*) values were calculated for the 138 grey matter regions (*Klein and Tourville, 2012*). The grey matter density was used in this study as a measure of structural atrophy.

## Tau PET

A 370-MBq/kg bolus injection of tau-specific ligand $^{18}$F-AV-1451 ([F- 18] T807) was given to each subject, and 30-min (6 × 5 min frames) brain PET scans were acquired at 75 min after injection (N = 233). As previously described (*Jagust et al., 2010*), the images were preprocessed as follows: (1) dynamic co-registration, (2) across-time averaging, (3) resampling and reorientation from native space to a standard voxel image grid space ('AC-PC' space), and (4) using ion parameters obtained for the structural T1 image with the nearest acquisition date, all tau images were normalized to the MNI space (*Evans et al., 1994*). The cerebellum was used as a reference to calculate SUVR values for the 138 grey matter regions.

## Anatomical connectivity estimation

The connectivity matrix was constructed in DSI Studio (http://dsi-studio.labsolver.org) using a group average template from 1065 subjects (*Yeh et al., 2018*). A multi-shell high-angular-resolution diffusion scheme was used, and the b-values were 990, 1985, and 2980 s/mm$^2$. The total number of sampling directions was 270. The in-plane resolution and slice thickness were 1.25 mm. The diffusion data were reconstructed in the MNI space using q-space diffeomorphic reconstruction to obtain the spin distribution function (*Yeh and Tseng, 2011*; *Yeh et al., 2010*). The sampling length and output resolution were set to 2.5 and 1 mm, respectively. The restricted diffusion was quantified using restricted diffusion imaging (*Yeh et al., 2017*) and a deterministic fibre tracking algorithm was used (*Yeh et al., 2013*). Using the brain atlas previously described under 'Methods' ('Whole-brain gene expression data and brain parcellation'), seeding was placed on the whole brain while setting the QA threshold to 0.15. The angular threshold was randomly varied from 15 to 90 degrees and the step size from 0.5 to 1.5 voxels. The fibre trajectories were smoothed by averaging the propagation direction with a percentage of the previous direction, which was randomly selected from 0 to 95%. Tracks with lengths shorter than 30 mm or longer than 300 mm were then discarded. A total of 100,000 tracts were calculated, and the connectivity matrix was obtained by using count of the connecting tracks.

## Gene Expression Multifactorial Causal Model

In the basic MCM formulation (*Iturria-Medina et al., 2017*), the brain is considered as a dynamic multifactorial causal system, in which (i) each variable represents a relevant macroscopic biological factor at a given brain region (e.g., tau and amyloid proteins, CBF, neuronal activity at rest, grey matter density) and (ii) alterations in each biological factor are caused by direct factor-factor interactions, the intra-brain propagation of factor-specific alterations (e.g., tau and amyloid spreading), and external inputs (e.g., drugs). Here, we extend this approach to incorporate GE at the regional level. Specifically, we examine how macroscopic biological alterations at each brain region, and the associated macroscopic factor-factor interactions, are controlled by the regional genetic activity.

The GE-MCM is therefore defined by (i) the influence of each gene on the local direct interactions among all the macroscopic factors, constrained within each brain region, and (ii) the potential spreading of macroscopic factor-specific alterations through anatomical and/or vascular networks. Mathematically, these processes can be described as

$$\frac{dS_i^m}{dt} = \sum_{n=1}^{N_{factors}} \left( \alpha_o^{n \to m} + \sum_{k=1}^{N_{genes}} \alpha_k^{n \to m} G_i^k \right) S_i^n + \sum_{\substack{j=1 \\ j \neq i}}^{N_{rois}} C_{ji}^m \left( S_j^m - S_i^m \right) S^m \tag{1}$$

where $N_{genes} = 976$ is the number of genes. Each gene was normalized by z-score across $N_{rois} = 138$ brain grey matter regions of interest (a gene $i$ is denoted as $G_i$; for region names, see *Supplementary file 3*). $N_{factors} = 6$ is the number of biological factors measured at the same brain regions (i.e., Aβ deposition, tau deposition, CBF, glucose metabolism, functional activity at rest, and grey matter density). Each node, corresponding to a given biological factor $m$ and region $i$, is characterized by $S_i^m \in R$.

In the equation, $\frac{dS_i^m}{dt}$ is the local longitudinal alteration of a macroscopic factor $m$ at the region $i$, because of the foregoing multiscale interactions. The first term on the right models the local direct influences of multiple macroscopic biological factors on the given factor $m$. The interaction

parameters ($\alpha_o^{n \to m}, \alpha_k^{n \to m}$) and gene expression ($G_i^k$) modulate the direct within-region impact of the factor $n$ on $m$, including intra-factor effects, that is, when $n = m$. $\sum_{\substack{j=1 \\ j \neq i}}^{N_{rois}} C_{ji}^m \left( S_j^m - S_i^m \right) S^m$ reflects the resultant signal propagation of factor $m$ from region $i$ to other brain regions through the physical network $C_{ji}^m$.

The GE-MCM model can advance our mechanistic understanding of the complex processes of aging and neurodegeneration. Its ability to map a healthy gene expression template to each subject allows us to model how the spatial distribution of transcriptome drives the multifactorial alteration observed in the brain. The interaction parameter $\alpha_k^{n \to m}$ is an implicit quantitative measure of dysregulation or deviation of gene expression from normal patterns. By fitting the model at the individual level, it is possible to identify subject-specific genetic targets for personalized treatments of AD and enhancing healthy aging.

## Model evaluation

The GE-MCM differential equation (1) was solved for each participant. For each subject $j$ and biological factor $m$, $\frac{dS_i^m(j)}{dt}$ was calculated between each pair of consecutive time points, and the regional values obtained were concatenated into a subject- and factor-specific vector ($\frac{dS^m(j)}{dt}$) with $N_{rois} \cdot (N_{times} - 1)$ unique values. This concatenation allowed us to express the evaluation of the model parameters ($\alpha_o^{n \to m}, \alpha_k^{n \to m}$) as a regression problem (with $\frac{dS^m(j)}{dt}$ as the dependent variable). We applied a Bayesian sparse linear regression with horseshoe hierarchy to identify the distribution of the model parameters (*Carvalho et al., 2010*; *Makalic and Schmidt, 2016*). Due to the high dimensionality of the data, a computationally efficient algorithm was used to sample the posterior Gaussian distribution of the regression coefficients (*Bhattacharya et al., 2016*), and the algorithm was implemented in MATLAB (*Makalic and Schmidt, 2016*). Through Markov chain Monte Carlo, we generated 500 samples of each regression coefficient after discarding the first 1000 burn-in simulations. All 500 samples were averaged, and 5863 coefficients were obtained for every subject and biological factor. For subsequent analysis, we used 5856 coefficients (transcriptomic-imaging parameters) that corresponded to the measure of transcriptomic effect on the interaction of a macroscopic imaging-based factor with the other macroscopic factors, in driving a longitudinal biological factor alteration.

Next, we sought to identify the top genes mediating cognitive and behavioural changes in healthy aging and AD progression. First, we identified 113 clinically stable HC subjects who did not convert to MCI or AD stage within 7.8 ± 2.9 years. In addition, we selected 129 diseased subjects diagnosed with AD at baseline or AD converters (i.e., HC and MCI subjects that advanced to AD within 3.7 ± 2.9 years). For each independent subset of subjects (i.e., stable HC or diseased subjects), we combined the transcriptomic-imaging parameters across the six longitudinal biological factor alterations (*Figure 1D*). We then evaluated the across-population stability of these model parameters via their 99% confidence intervals (99% CI). Next, the rate of change of cognitive scores was calculated for each subject (7.2 ± 2.6 time points for HC and 6.3 ± 3.0 time points for AD). We applied SVD multivariate analysis to evaluate how the stable transcriptomic-imaging interactions mediate group-specific changes in cognitive/clinical scores (age-related slopes of MMSE, ADAS-11, ADAS-13, EM, and EF). For each group (i.e., HC or AD), SVD identified a few pairs of 'principal components' that maximize the cross-correlation between the two sets of variables (*Carbonell et al., 2020*; *Worsley et al., 2005*). Then it mapped the gene-imaging parameters onto the obtained PCs. This mapping provides the score (or contribution) of a gene-imaging parameter to a PC. Next, the significant PCs were identified by running 10,000 permutations. To identify the genes (gene-imaging parameters) with large and reliable contributions on the significant PC, we drew 10,000 bootstrap samples and calculated the bootstrap ratio of the gene-imaging parameters. The bootstrap ratio is obtained by dividing the gene-imaging saliences (contributions) by their respective bootstrap standard errors. It allowed us to assess the reliability of the genetic contributions (*McIntosh and Lobaugh, 2004*). Hence, top aging- or AD-related causal genes were identified by selecting the parameters with a bootstrap ratio above 2.58, which is approximately equivalent to a z-score for 99% CI if the bootstrap distribution is normal (*Efron and Tibshirani, 1986*).

## Code availability

We anticipate that the GE-MCM will be released soon as part of an already available open-access user-friendly multi-tool software for researchers (*Iturria-Medina et al., 2021*) at https://www.neuropm-lab.com/neuropm-box.html. Importantly, standalone applications for Linux, macOS, and Windows systems are provided (MATLAB license and/or programming expertise are not required).

## Acknowledgements

This research was undertaken by funding from the *Canada First Research Excellence Fund*, awarded to McGill University for the *Healthy Brains for Healthy Lives Initiative*, the *Fonds de la Recherche en Sante du Quebec* (FRQS) Research Scholars Junior 1, the Canada Research Chair Tier-2 and the Weston Brain Institute Rapid Response programs 2018 and 2019 awards to YIM, and the *Brain Canada Foundation* and *Health Canada* support to the McConnell Brain Imaging Centre at the Montreal Neurological Institute. Dataset-1 collection and sharing was provided by Allen Institute for Brain Science (funded in part by the Department of Health and Human Services Health Resources and Services Administration awards 1C76HF15069-01-00 and 1C76HF19619-01-00). The Allen Human Brain Atlas is also supported by Paul G Allen, Mr. and Mrs. Richard Daly, Mr. and Mrs. Peter Eschenbach, Mr. William E Fay, Jr., Mr. Nathan Hansen, Mr. and Mrs. Richard Jernstedt, Bruce and Kathryn Johnson, Mr. and Mrs. Martin Koldyke, Michael and Suzanne Moskow, Bruce and Gwill Newman, Mr. and Mrs. Peter Pond, Mr. and Mrs. James N Bayer, Jr., Mr. and Mrs. Richard L Joutras, Mr. and Mrs. Robert Lorch, Lee and Jeanne Zehrer, and the board members of Brain Research Foundation.

The Dataset-2 collection and sharing for this project was funded by ADNI (National Institutes of Health Grant U01 AG024904) and DOD ADNI (Department of Defense award W81XWH-12-2-0012). ADNI is funded by the National Institute on Aging, the National Institute of Biomedical Imaging and Bioengineering, and through generous contributions from the following: AbbVie, Alzheimer's Association; Alzheimer's Drug Discovery Foundation; Araclon Biotech; BioClinica, Inc; Biogen; Bristol-Myers Squibb Company; CereSpir, Inc; Eisai, Inc; Elan Pharmaceuticals, Inc; Eli Lilly and Company; EuroImmun; F Hoffmann-La Roche Ltd and its affiliated company Genentech, Inc; Fujirebio; GE Healthcare; IXICO Ltd; Janssen Alzheimer Immunotherapy Research and Development, LLC; Johnson and Johnson Pharmaceutical Research and Development, LLC; Lumosity; Lundbeck; Merck and Co, Inc; Meso Scale Diagnostics, LLC; NeuroRx Research; Neurotrack Technologies; Novartis Pharmaceuticals Corporation; Pfizer, Inc; Piramal Imaging; Servier; Takeda Pharmaceutical Company; and Transition Therapeutics. The Canadian Institutes of Health Research is providing funds to support ADNI clinical sites in Canada. Private sector contributions are facilitated by the Foundation for the National Institutes of Health (http://www.fnih.org/). The grantee organization is the Northern California Institute for Research and Education, and the study is coordinated by the Alzheimer's Disease Cooperative Study at the University of California, San Diego. ADNI data is disseminated by the Laboratory for Neuro Imaging at the University of Southern California.

## Additional information

### Competing interests

Felix Carbonell: Felix Carbonell is affiliated with Biospective Inc. The author has no financial interests to declare. The other authors declare that no competing interests exist.

### Funding

| Funder | Grant reference number | Author |
|---|---|---|
| Canada First Research Excellence Fund | Healthy Brains for Healthy Lives Initiative | Yasser Iturria-Medina |
| Fonds de Recherche du Québec - Santé | Research Scholars Junior 1 | Yasser Iturria-Medina |
| Canada Research Chairs | Tier-2 | Yasser Iturria-Medina |
| Weston Brain Institute | Rapid Response: Canada | Yasser Iturria-Medina |

| | | |
|---|---|---|
| | | 2018 and 2019 |
| Health Canada | Canada Brain Research Fund | Yasser Iturria-Medina |
| National Institutes of Health | U01 AG024904 | Alzheimer's Disease Neuroimaging Initiative |
| Department of Defense | W81XWH-12-2-0012 | Alzheimer's Disease Neuroimaging Initiative |

The funders had no role in study design, data collection and interpretation, or the decision to submit the work for publication.

### Author contributions

Quadri Adewale, Conceptualization, Resources, Data curation, Software, Formal analysis, Validation, Visualization, Methodology, Writing - original draft, Writing - review and editing, Result interpretation; Ahmed F Khan, Writing - review and editing, Result interpretation; Felix Carbonell, Software, Writing - review and editing, Result interpretation; Yasser Iturria-Medina, Conceptualization, Resources, Supervision, Funding acquisition, Methodology, Writing - review and editing, Result interpretation; Alzheimer's Disease Neuroimaging Initiative, Data generation

### Author ORCIDs

Quadri Adewale ⓘ https://orcid.org/0000-0001-5090-6140
Yasser Iturria-Medina ⓘ https://orcid.org/0000-0002-9345-0347

### Ethics

Human subjects: This article does not contain any studies with human participants performed by any of the authors. The authors obtained approval from the ADNI Data Sharing and Publications Committee for data use and publication. As per ADNI protocols, the study was conducted according to Good Clinical Practice guidelines, the Declaration of Helsinki, US 21CFR Part 50 - Protection of Human Subjects, and Part 56 - Institutional Review Boards, and pursuant to state and federal HIPAA regulations (adni.loni.usc.edu). Study subjects and/or authorized representatives gave written informed consent at the time of enrollment for sample collection and completed questionnaires approved by each participating site Institutional Review Board (IRB).

### Decision letter and Author response

Decision letter https://doi.org/10.7554/eLife.62589.sa1
Author response https://doi.org/10.7554/eLife.62589.sa2

## Additional files

### Supplementary files

- Source code 1. Codes for statistical analyses and plotting.
- Supplementary file 1. Main demographic characteristics of the included ADNI subjects.
- Supplementary file 2. List of 976 genes used in this study.
- Supplementary file 3. Brain regions used in this study.
- Supplementary file 4. Distribution of stable gene-imaging interaction parameters in healthy aging and AD progression (99% CI).
- Supplementary file 5. Identified molecular pathways underlying AD progression.
- Transparent reporting form

### Data availability

All data used in this study are publicly available at the Allen Human Brain Atlas website (Hawrylycz et al., 2012. Nature, 489:391-399; http://human.brain-map.org/static/download) and the Alzheimer's Disease Neuroimaging Initiative (ADNI) database (Peterson et al., 2010. Neurology, 74(3): 201-209;

http://adni.loni.usc.edu/data-samples/access-data/). While AHBA data do not require any registration for download, ADNI data can be accessed by creating an account and submitting an online application form. The application includes the investigator's institutional affiliation and the proposed uses of the ADNI data (scientific investigation, teaching, or planning clinical research studies). ADNI data may not be used for commercial products or redistributed in any way.

The following previously published datasets were used:

| Author(s) | Year | Dataset title | Dataset URL | Database and Identifier |
|---|---|---|---|---|
| Hawrylycz et al. | 2012 | The Allen Human Brain Atlas | http://human.brain-map.org/static/download | Allen Human Brain Atlas, RRID:SCR_007416 |
| Petersen et al. | 2010 | The Alzheimer's Disease Neuroimaging Initiative | http://adni.loni.usc.edu/data-samples/access-data/ | Alzheimer's Disease Neuroimaging Initiative, RRID:SCR_003007 |

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
