## [Decision Letter]

**Acceptance summary:**

Both aging and Alzheimer's disease (AD) are complex traits that involve several different biological aberrations occurring at both macro- and microscopic levels. However, few studies have linked the macroscopic brain changes that occur in aging and AD to molecular changes. This paper integrates regional gene expression patterns to the authors previously developed multifactorial causal model (MCM), to identify potentially causal genes and pathways driving neuropathological progression. The novelty of this paper lies in the insights it provides into the interaction between genes and macroscopic factors in aging / AD and possible mechanistic roles of the identified genetic determinants. We all felt that the overall aim of the work is valuable, to investigate relationships between gene expression, brain imaging modalities and cognition and that the resulting findings are important.

**Decision letter after peer review:**

Thank you for submitting your article "Integrated transcriptomic and neuroimaging brain model decodes biological mechanisms in aging and Alzheimer's disease" for consideration by *eLife*. Your article has been reviewed by 2 peer reviewers, and the evaluation has been overseen by a Reviewing Editor and Jessica Tyler as the Senior Editor. The reviewers have opted to remain anonymous.

The reviewers have discussed the reviews with one another and the Reviewing Editor has drafted this decision to help you prepare a revised submission.

Both aging and Alzheimer's disease (AD) are complex traits that involve several different biological aberrations occurring at both macro- and microscopic levels. However, few studies have linked the macroscopic brain changes that occur in aging and AD to molecular changes. In this paper, Adewale et al. integrated regional gene expression pattern to their previously developed multifactorial causal model (MCM) to identify potentially causal genes and pathways driving neuropathological progression. The novelty of this paper lies in the insights it provides into the interaction between genes and macroscopic factors in aging / AD and possible mechanistic roles of the identified genetic determinants. We all felt that the overall aim of the work is valuable, to investigate relationships between gene expression, brain imaging modalities and cognition. A few areas require clarification. Also, the manuscript is rather disorganized and there are a number of methodological inconsistencies. The text is not well organized into clearly differentiated parts and there are lots of overlaps: for example the Introduction contains methodological descriptions, the results contain discussion aspects, etc. The paper would also benefit from language editing. The comments below can mainly be addressed by rewriting.

Essential revisions:

1. Abstract. The abstract doesn't reflect well the results of the study. The writing is unclear: for example, what is the difference between "spatial biological scales" (e.g. genetic or brain imaging) and "macroscopic biological factors" (e.g. functional activity or atrophy)? In particular, atrophy and functional activity are measured with brain imaging so this means that atrophy and functional activity would be both "macroscopic biological factors" and "spatial biological scales"? The abstract doesn't mention cognition or cognitive decline, even though these aspects were part of the analyses. The last sentence of the abstract is vague: "This model offers novel mechanistic insights into the multiscale brain reorganization in aging and neurodegeneration, with implications for identifying effective genetic targets for extending healthy aging and AD treatment"; in particular, it is unclear that the study gives insights into "multiscale brain reorganization", and also the paper doesn't present any results on "brain reorganization". Finally, the implications of the paper to "further our physiological understanding of healthy aging and neurodegeneration" are unclear to me.

2. Introduction. Regarding the sentence: "However, in both aging and disease research, most studies are performed at either micro or macroscopic scale,.." not entirely clear what authors mean by "micro" and "macro", and in general many terms are presented without clear definitions. It is also written that: "As a result, we continue to lack brain generative models integrating a large set of genetic activities with multimodal brain properties and cognitive/clinical integrity." But the generative model proposed under methods does not seem to include cognitive or clinical variables. Also what is the meaning of "clinical integrity"?

3. Results. This section includes a mixture of material that would better fit within Introduction, Methods or Discussion, so it is written in a confusing way. How the genes were selected is unclear, for example "The remaining 1.5% (15 genes) were selected from the list of AD-associated genes provided by GeneCards" (it seems arbitrary to select some genes from one database and some other genes from other database, the rationale is not clear). it is then written: "Hence, we sought to identify causal genes underlying longitudinal cognitive changes in healthy aging." It is not clear how the longitudinal cognitive changes were calculated, and based on how many longitudinal time points?

4. Results, on page 6, it is written that "All subjects presented at least four imaging modalities and three longitudinal imaging evaluations". However, the integrative model presented in the methods should incorporate 6 imaging modalities, so there seems to be an inconsistency between methods and the results.

5. Figures 3 and 4. It is not clear why in Figure 3 (healthy aging) there are only 3 longitudinal factor alterations, why there are 6 longitudinal factor alterations in Figure 4 (Alzheimer's disease spectrum). Also, it is unclear why only the genes contributing to the first principal component were discussed in the paper, given that in healthy aging the first component contributed only to 50% of the variance (so what is the role of the other components?), and why the results were so different in AD where the first component was responsible for most of the variance?

6. Methods. About study participants: "This study involved 944 individuals with multimodal brain imaging from the Alzheimer's Disease Neuroimaging Initiative (ADNI)", but then only n=460 participants were used to test the method. How were these 460 selected? This text is confusing because it is not clear how the whole sample of 944 individuals was used in the analyses. Also there are unequal numbers of subjects with different imaging modalities: n=337 for ASL, n=912 for amyloid PET, n=186 for resting-state fMRI, n=821 for FDG PET, not clear how many subjects had MRI scans, and unclear how many of all the mentioned scans were baseline and/or longitudinal. Also, what atlas was used to extract data on 138 brain regions?

7. Methods, regarding the Gene Expression Multifactorial Causal Model (GE-MCM)

Authors selected 990 genes which were normalised across 𝑁_𝑟𝑜𝑖𝑠_ = 138 brain regions of interest covering most of the brain's grey matter. Not clear why this number of genes, how selected, and then how where they "normalised"? In Equation 1, there were different numbers of subjects for each image modality, so how was the information combined, and what possible biases may have affected the predictions from the model (based on the very different amount of imaging data available for each of the 6 imaging modalities)? How were the mentioned anatomical and/or vascular networks calculated? Why were only 5 youngest healthy subjects from ADNI selected as control, as this sounds like a very small group to be a control?

8. The authors chose ~900 landmark genes that were previously shown to cover most information in human transcriptome and 15 additional genes from a list of AD-associated genes provided by GeneCards. How are the 15 genes chosen from the AD list? Why is it that some of the key genes associated with AD, eg APOE, is not included in the 15 AD-related genes when it is also not in the landmark genes? Similarly, I noticed that many of the aging-associated genes (eg FOXO, SOD etc) are not included in the analysis. Would it be more informative to include aging-related genes (from database like GenAge) for the analysis, or can the authors explain the rationale for not doing so?

9. The GE-MCM differential equation has a time component, but the gene component (Gi) in the equation is assumed to be static as it is derived from the standardized AHBA microarray data. Does this assumption hold? It is also stated that the AHBA mRNA expression data are obtained from samples from 6 adult brain, which is presumably a different age group than the ADNI subjects. While I understand it is beyond the scope of this paper to obtain longitudinal individual genetic expression data, the authors should explain these caveats / limitations in the main text.

10. Why is the number of stable gene-imaging parameters in healthy aging much lower than AD progression (51 vs ~1000)?

11. It will be more insightful if the authors can discuss what the results say about normal aging vs AD in the discussion (eg. How is aging and AD related (or not), or compare the changes driving aging-disease continuum) as these two sections (aging vs AD) currently read like separate, unrelated sections.

---

## [Author Response]

[…] We all felt that the overall aim of the work is valuable, to investigate relationships between gene expression, brain imaging modalities and cognition. A few areas require clarification. Also, the manuscript is rather disorganized and there are a number of methodological inconsistencies. The text is not well organized into clearly differentiated parts and there are lots of overlaps: for example the Introduction contains methodological descriptions, the results contain discussion aspects, etc. The paper would also benefit from language editing. The comments below can mainly be addressed by rewriting.

We sincerely appreciate the reviewers and the editors for their comments and suggestions in helping to substantially improve this manuscript. Motivated by these, we have made multiple modifications to our manuscript (highlighted in yellow in the revised version), with major changes including:

– We restricted our analysis to 976 landmark genes (with leading roles in central biological functions) and removed the 15 additional genes from GeneCards (see Reply to Comment 3). Based on the latest (and recently corrected) list of landmark genes updated on the Connectivity Map (CMap) (Subramanian et al., 2017), 193 genes were updated/replaced in the list of landmark genes used in our initial manuscript. Hence, to avoid the ‘arbitrary’ selection of additional genes, the analyses in this revised manuscript were performed with the corrected CMap list of landmark genes only (see “Capturing Gene and Macroscopic Factor Interactions in the Human Brain” subsection, Results).

– We divided the Discussion section into three main subsections. The first and second subsections are new (focused on interpreting the multifactorial interactions observed in our results, and comparing the results obtained in healthy aging analyses with those of AD), while the third section is a subtle modification of the initial Discussion section. The first, second and third subsections are titled “Gene Expression Patterns Modulate Multifactorial Interactions in Healthy Aging and AD Progression”, “Aging and Alzheimer’s Disease Have Both Common and Distinct Mechanisms”, and “Towards a Genetic Approach to Extending Healthy Aging and Treating Alzheimer’s Disease”, respectively.

Essential revisions:1. Abstract. The abstract doesn't reflect well the results of the study. The writing is unclear: for example, what is the difference between "spatial biological scales" (e.g. genetic or brain imaging) and "macroscopic biological factors" (e.g. functional activity or atrophy)? In particular, atrophy and functional activity are measured with brain imaging so this means that atrophy and functional activity would be both "macroscopic biological factors" and "spatial biological scales"? The abstract doesn't mention cognition or cognitive decline, even though these aspects were part of the analyses. The last sentence of the abstract is vague: "This model offers novel mechanistic insights into the multiscale brain reorganization in aging and neurodegeneration, with implications for identifying effective genetic targets for extending healthy aging and AD treatment"; in particular, it is unclear that the study gives insights into "multiscale brain reorganization", and also the paper doesn't present any results on "brain reorganization". Finally, the implications of the paper to "further our physiological understanding of healthy aging and neurodegeneration" are unclear to me.

We thank the reviewer for this comment.

i. Motivated by this, we have rewritten the abstract thus:

“Both healthy aging and Alzheimer’s disease (AD) are characterized by concurrent alterations in several biological factors. […] Overall, this personalized model offers novel insights into the multiscale alterations in the elderly brain, with important implications for identifying effective genetic targets for extending healthy aging and treating AD progression.”

**ii. By multiscale brain reorganization, we mean multiscale brain changes/alterations. Our model analyzed changes in amyloid, tau, brain structure, functional activity, metabolism and blood flow across times and regions, and thus characterized brain reorganization. We understand using the reorganization might be ambiguous, hence we have changed it to “alterations”.**

**iii. We agree with the reviewer that it is not clear how the study advance physiological understanding. We have removed the word “physiological”.**

2. Introduction. Regarding the sentence: "However, in both aging and disease research, most studies are performed at either micro or macroscopic scale,.." not entirely clear what authors mean by "micro" and "macro", and in general many terms are presented without clear definitions. It is also written that: "As a result, we continue to lack brain generative models integrating a large set of genetic activities with multimodal brain properties and cognitive/clinical integrity." But the generative model proposed under methods does not seem to include cognitive or clinical variables. Also what is the meaning of "clinical integrity"?

We thank the reviewer for this comment.

i. By microscopic and macroscopic scales, we mean order of ~10^-6^ m and ~10^-2^ m, respectively. We have updated the new manuscript to reflect this definition. A new paragraph under Introduction section reads:

“Indeed, at the microscopic scale (~10^-6^ m), transcriptomics and proteomics analysis of the brain have paved the way for deciphering the mechanistic underpinnings of healthy aging and AD (Dillman et al., 2017; Iturria-Medina et al., 2020; E. C. B. Johnson et al., 2020; Mostafavi et al., 2018; Tanaka et al., 2018). […] However, in both aging and disease research, most studies incorporate brain measurements at either micro (e.g., transcriptomics) or macroscopic scale (e.g., PET imaging), failing to detect the direct causal relationships between several biologically factors at multiple spatial resolutions.”

ii. Yes, we agree with the reviewer that our generative model does not intrinsically include cognitive variables, even though, we investigate how the obtained model parameters are associated clinical variables. We have modified the previous sentence, removing “… and cognitive/clinical integrity”.

3. Results. This section includes a mixture of material that would better fit within Introduction, Methods or Discussion, so it is written in a confusing way. How the genes were selected is unclear, for example "The remaining 1.5% (15 genes) were selected from the list of AD-associated genes provided by GeneCards" (it seems arbitrary to select some genes from one database and some other genes from other database, the rationale is not clear). it is then written: "Hence, we sought to identify causal genes underlying longitudinal cognitive changes in healthy aging." It is not clear how the longitudinal cognitive changes were calculated, and based on how many longitudinal time points?

Thank you for this comment.

i. Driven by it, we have restricted the new Results section to only the necessary details. Specifically, the results are not discussed from a biological perspective (which is now done in the Discussion section).

ii. We agree with the reviewer that the selection of the additional genes was arbitrary. Initially, we included 15 additional genes to see if our model would identify well known AD genes. However, considering that we also performed healthy aging analysis without additional aging-related genes, and the selection of both aging- and AD-related genes may be performed in several different ways (ranging from a few to hundreds or thousands of previously cited genes), we have excluded the 15 additional AD-related genes. Subsequently, we have repeated our analysis using just the CMap’s 976 landmark genes (Subramanian et al., 2017), which correspond to the topmost informative transcripts with the capacity to cover most of the information in the whole human transcriptome across a diversity of tissue types.

iii. Accordingly, we have updated the Results (subsection Identifying Genes Driving Biological and Cognitive Changes in Healthy Aging) as follows:

**“**Hence, we sought to identify causal genes underlying longitudinal cognitive changes in healthy aging. […] The cognitive changes were obtained as the age-related slopes of MMSE, ADAS, executive function (EF), and memory score (MEM) over an average of 7.2 time points (SD=2.6).”

**The Methods section have also been updated to include the details of time point and cognitive changes.**

4. Results, on page 6, it is written that "All subjects presented at least four imaging modalities and three longitudinal imaging evaluations". However, the integrative model presented in the methods should incorporate 6 imaging modalities, so there seems to be an inconsistency between methods and the results.

We thank the reviewer for pointing this out. We have removed the statement from the manuscript. Indeed, all subjects used for analyses have 6 neuroimaging modalities, after imputation. However, the imputation was only performed for subjects with at least four neuroimaging modalities and three longitudinal imaging evaluations.

5. Figures 3 and 4. It is not clear why in Figure 3 (healthy aging) there are only 3 longitudinal factor alterations, why there are 6 longitudinal factor alterations in Figure 4 (Alzheimer's disease spectrum). Also, it is unclear why only the genes contributing to the first principal component were discussed in the paper, given that in healthy aging the first component contributed only to 50% of the variance (so what is the role of the other components?), and why the results were so different in AD where the first component was responsible for most of the variance?

Thank you for giving us the opportunity to improve the paper through comment.

i. Figure 3 has three longitudinal factor alterations because only those three factors had significant alterations with healthy aging, even though we did analysis on all the 6 factors. We added a new subsection (Aging and Alzheimer’s Disease Have Common and Distinct Mechanisms) under Discussion, with a paragraph thus:

“At the individual level, the fitted gene-imaging parameters are assumed to reflect the gene-specific deformations required to fit the data. […] We attribute the greater number of significant parameters in AD to more genetic dysregulations and biological mechanism alterations in the disorder (Iturria-Medina et al., 2020; Mostafavi et al., 2018).”

ii. Please notice that we only discussed the genes contributing to the first principal component (PC) because only the first PC was significant after running permutation analysis. We have updated Figures 3A and 4A to show the p-value of each PC.

iii. Thank you for spotting this. There was a large difference in the variances explained in the aging and AD analyses because for the AD analysis we only used the baseline cognitive scores instead of the age-related cognitive slopes (as for the healthy population). For consistency, in the revised manuscript version, we have used age-related cognitive slopes for both healthy aging and AD, and the variance explained are not very disparate (50% versus 63%).

6. Methods. About study participants: "This study involved 944 individuals with multimodal brain imaging from the Alzheimer's Disease Neuroimaging Initiative (ADNI)", but then only n=460 participants were used to test the method. How were these 460 selected? This text is confusing because it is not clear how the whole sample of 944 individuals was used in the analyses. Also there are unequal numbers of subjects with different imaging modalities: n=337 for ASL, n=912 for amyloid PET, n=186 for resting-state fMRI, n=821 for FDG PET, not clear how many subjects had MRI scans, and unclear how many of all the mentioned scans were baseline and/or longitudinal. Also, what atlas was used to extract data on 138 brain regions?

We sincerely appreciate the reviewer’s comment, which could help to considerably enhance the extension and comprehension of our research.

i. Regarding the number of subjects available for each imaging modality, we extended the associated details and included them in the *Materials and methods* section, *Data Description and Processing* subsection, *Study Participants* subsubsection. Specifically, we added the following paragraph:

“This study involved 944 individuals with six multimodal brain imaging from the Alzheimer’s Disease Neuroimaging Initiative (ADNI) (adni.loni.usc.edu; see Supplementary Figure S1). […] Among the 460 participants, 151 were clinically identified as asymptomatic or healthy control (HC), 161 with early mild cognitive impairment (EMCI), 113 with late mild cognitive impairment (LMCI) and 35 with probable Alzheimer’s disease (AD).”

ii. We have also updated the distribution of the imaging modalities (before imputation) across the 460 subjects under the Materials and methods section, Multimodal Imaging Modalities subsection. Specifically, we have N=460 for MRI, N=213 for ASL, N=459 for amyloid PET, N=233 for tau, N=148 for resting-state fMRI, and N=455 for FDG PET.

iii. The details of the atlas used can be found under Methods: Whole-Brain Gene Expression Data and Brain Parcellation, as follows:

“The brain was parcellated into 144 grey matter regions and the average expression value of each gene was calculated for each region. […] The remaining 138 regions were used for analyses (see Supplementary Table S3).”

7. Methods, regarding the Gene Expression Multifactorial Causal Model (GE-MCM)Authors selected 990 genes which were normalised across 𝑁_𝑟𝑜𝑖𝑠_ = 138 brain regions of interest covering most of the brain's grey matter. Not clear why this number of genes, how selected, and then how where they "normalised"? In Equation 1, there were different numbers of subjects for each image modality, so how was the information combined, and what possible biases may have affected the predictions from the model (based on the very different amount of imaging data available for each of the 6 imaging modalities)? How were the mentioned anatomical and/or vascular networks calculated? Why were only 5 youngest healthy subjects from ADNI selected as control, as this sounds like a very small group to be a control?

We thank the reviewer for the opportunity to clarify these details.

i. With respect to the number of genes, it is infeasible for us to use thousands of genes for our analysis due to the limited number of subjects. Hence, we selected the ~1000 landmark identified by (Subramanian et al., 2017) as these genes were shown to consistently cover most of the information of the whole human transcriptome.

ii. Each gene’s expression level was normalized by calculating the z-score across the 138 brain regions (mean 0, std 1). We have updated the Methods section, Gene Expression Multifactorial Causal Model subsection as follows:

“Ngenes=976 is the number of genes. Each gene was normalized by z-score across Nrois=138 brain regions of interest covering most of the brain's grey matter (each gene *i* is denoted as Gi).”

iii. Yes, the reviewer is right that there were different number of subjects for each modality. However, after imputation, all subjects had completed all the 6 neuroimaging modalities. We have extended the associated details and included them in the *Materials and methods* section, *Data Description and Processing* subsection, *Study Participants* subsubsection. The details can be found under the reply (i) to Comment 6 above.

iv. We have added a new subsection titled *Anatomical Connectivity Estimation* to the *Methods* section. The subsection reads as follows:

“The connectivity matrix was constructed in DSI Studio (http://dsi-studio.labsolver.org) using a group average template from 1065 subjects (Yeh et al., 2018). […] A total of 100000 tracts were calculated and connectivity matrix was obtained by using count of the connecting tracks.”

v. The 5 youngest subjects were only used to evaluate baseline values of each imaging modality to obtain the deviations of the modalities from an initial or a pathology free state. In practice, this subtraction was indeed symbolic as it is not considered in the model’s optimization, having no effect on the longitudinal alteration (rate of change) of each modality or on the obtained model parameters. For simplicity (and to avoid any associated confusion), we have removed this statement from the manuscript.

8. The authors chose ~900 landmark genes that were previously shown to cover most information in human transcriptome and 15 additional genes from a list of AD-associated genes provided by GeneCards. How are the 15 genes chosen from the AD list? Why is it that some of the key genes associated with AD, eg APOE, is not included in the 15 AD-related genes when it is also not in the landmark genes? Similarly, I noticed that many of the aging-associated genes (eg FOXO, SOD etc) are not included in the analysis. Would it be more informative to include aging-related genes (from database like GenAge) for the analysis, or can the authors explain the rationale for not doing so?

A very insightful comment from the reviewer. Please see Reply to Comment 3. As mentioned, our initial selection of 15 additional genes was arbitrary based on frequently reported genes with differential expression pattern in AD. We have however excluded these additional 15 genes, and repeated our analysis based on the landmark genes only.

9. The GE-MCM differential equation has a time component, but the gene component (Gi) in the equation is assumed to be static as it is derived from the standardized AHBA microarray data. Does this assumption hold? It is also stated that the AHBA mRNA expression data are obtained from samples from 6 adult brain, which is presumably a different age group than the ADNI subjects. While I understand it is beyond the scope of this paper to obtain longitudinal individual genetic expression data, the authors should explain these caveats / limitations in the main text.

We thank the reviewer for this comment as it will help in better clarifying the method.

i. Regarding the usage of static gene expression, under the Discussion section, we added a new subsection titled Aging and Alzheimer’s Disease Have Common and Distinct Mechanisms which has the following explanation:

“In this study, we used a single gene expression template for all the subjects due to the unavailability of individual whole-brain gene expression. However, notice that even though this template has spatial but no temporal variation, for each gene, a model parameter controls its interaction (at the individual level) with each time-varying neuroimaging modality (i.e. the estimated transcriptomic-imaging parameters). At the individual level, the fitted gene-imaging parameters are assumed to reflect the gene-specific deformations required to fit the data. Consequently, these parameters represent quantitative measures of the individual dysregulation or deviation in gene expression patterns, and when analyzed across the entire population (e.g. via SVD analysis), the parameters can be used to detect cognitive/clinical related genetic associations.”

ii. We agree with the reviewer, and we added the following to the Discussion section, Towards a Genetic Approach to Extending Healthy Aging and Treating Alzheimer’s Disease subsection.

“There is inherent bias in the merged gene expression data from AHBA due to individual variability, and the AHBA subjects are not very representative of the typical age range in ADNI cohort. […] Thus, our future work will therefore focus on using personalized gene expression data from blood samples.”

10. Why is the number of stable gene-imaging parameters in healthy aging much lower than AD progression (51 vs ~1000)?

We appreciate this interesting question. We have added a new subsection under Discussion and titled it “Aging and Alzheimer’s Disease Have Common and Distinct Mechanisms”. The reason for differences is number of stable parameters is explained as follows:

“Thus, under normal aging, the parameters obtained from the model optimization should be close to zero. […] We attribute the greater number of significant parameters in AD to more genetic dysregulations and biological mechanism alterations in the disorder (Iturria-Medina et al., 2020; Mostafavi et al., 2018).”

11. It will be more insightful if the authors can discuss what the results say about normal aging vs AD in the discussion (eg. How is aging and AD related (or not), or compare the changes driving aging-disease continuum) as these two sections (aging vs AD) currently read like separate, unrelated sections.

Thank you very much for this suggestion. Motivated by it, we added a new subsubsection in the Discussion section, as follows:

“Aging and Alzheimer’s Disease Have Common and Distinct Mechanisms.

In this study, we used a single gene expression template for all the subjects due to the unavailability of individual whole-brain gene expression. […] Although several studies have shown the implication of altered mRNA splicing in AD, the exact role of *LSM6* gene in AD warrants further investigation (Johnson et al., 2018; Koch, 2018; Twine, Janitz, Wilkins, and Janitz, 2011).”

References:

https://doi.org/10.1016/j.chemolab.2016.03.019

Budanov, A. V., Lee, J. H., and Karin, M. (2010). Stressin' Sestrins take an aging fight. EMBO Mol Med, 2(10), 388-400. doi:10.1002/emmm.201000097.

Folch-Fortuny, A., Arteaga, F., and Ferrer, A. (2016). Missing Data Imputation Toolbox for MATLAB. Chemometrics and Intelligent Laboratory Systems, 154, 93-100. doi:

Iturria-Medina, Y., Khan, A. F., Adewale, Q., Shirazi, A. H., and Alzheimer's Disease Neuroimaging, I. (2020). Blood and brain gene expression trajectories mirror neuropathology and clinical deterioration in neurodegeneration. Brain, 143(2), 661-673. doi:10.1093/brain/awz400.

Iturria-Medina, Y., Sotero, R. C., Toussaint, P. J., Mateos-Perez, J. M., Evans, A. C., and Alzheimer's Disease Neuroimaging, I. (2016). Early role of vascular dysregulation on late-onset Alzheimer's disease based on multifactorial data-driven analysis. Nat Commun, 7, 11934. doi:10.1038/ncomms11934.

Jasinska, A. J., Service, S., Choi, O. W., DeYoung, J., Grujic, O., Kong, S. Y.,... Freimer, N. B. (2009). Identification of brain transcriptional variation reproduced in peripheral blood: an approach for mapping brain expression traits. Hum Mol Genet, 18(22), 4415-4427. doi:10.1093/hmg/ddp397.

Johnson, E. C. B., Dammer, E. B., Duong, D. M., Yin, L., Thambisetty, M., Troncoso, J. C.,... Seyfried, N. T. (2018). Deep proteomic network analysis of Alzheimer's disease brain reveals alterations in RNA binding proteins and RNA splicing associated with disease. Molecular neurodegeneration, 13(1), 52-52. doi:10.1186/s13024-018-0282-4.

Kim, M., Sujkowski, A., Namkoong, S., Gu, B., Cobb, T., Kim, B.,... Lee, J. H. (2020). Sestrins are evolutionarily conserved mediators of exercise benefits. Nature Communications, 11(1), 190. doi:10.1038/s41467-019-13442-5.

Koch, L. (2018). Altered splicing in Alzheimer transcriptomes. Nature Reviews Genetics, 19(12), 738-739. doi:10.1038/s41576-018-0064-4.

Mostafavi, S., Gaiteri, C., Sullivan, S. E., White, C. C., Tasaki, S., Xu, J.,... De Jager, P. L. (2018). A molecular network of the aging human brain provides insights into the pathology and cognitive decline of Alzheimer's disease. Nature Neuroscience, 21(6), 811-819. doi:10.1038/s41593-018-0154-9.

Palomero-Gallagher, N., and Zilles, K. (2019). Cortical layers: Cyto-, myelo-, receptor- and synaptic architecture in human cortical areas. Neuroimage, 197, 716-741. doi:10.1016/j.neuroimage.2017.08.035.

Subramanian, A., Narayan, R., Corsello, S. M., Peck, D. D., Natoli, T. E., Lu, X.,... Golub, T. R. (2017). A Next Generation Connectivity Map: L1000 Platform and the First 1,000,000 Profiles. Cell, 171(6), 1437-1452.e1417. doi:10.1016/j.cell.2017.10.049.

Sullivan, P. F., Fan, C., and Perou, C. M. (2006). Evaluating the comparability of gene expression in blood and brain. Am J Med Genet B Neuropsychiatr Genet, 141b(3), 261-268. doi:10.1002/ajmg.b.30272.

Toepper, M. (2017). Dissociating Normal Aging from Alzheimer's Disease: A View from Cognitive Neuroscience. Journal of Alzheimer's disease : JAD, 57(2), 331-352. doi:10.3233/JAD-161099

Twine, N. A., Janitz, K., Wilkins, M. R., and Janitz, M. (2011). Whole transcriptome sequencing reveals gene expression and splicing differences in brain regions affected by Alzheimer's disease. PLoS One, 6(1), e16266. doi:10.1371/journal.pone.0016266.

Ubaida-Mohien, C., Lyashkov, A., Gonzalez-Freire, M., Tharakan, R., Shardell, M., Moaddel, R.,... Ferrucci, L. (2019). Discovery proteomics in aging human skeletal muscle finds change in spliceosome, immunity, proteostasis and mitochondria. ELife, 8. doi:10.7554/eLife.49874.

Witt, S. H., Sommer, W. H., Hansson, A. C., Sticht, C., Rietschel, M., and Witt, C. C. (2013). Comparison of gene expression profiles in the blood, hippocampus and prefrontal cortex of rats. In Silico Pharmacology, 1(1), 15. doi:10.1186/2193-9616-1-15.

Xia, X., Jiang, Q., McDermott, J., and Han, J.-D. J. (2018). Aging and Alzheimer's disease: Comparison and associations from molecular to system level. Aging cell, 17(5), e12802-e12802. doi:10.1111/acel.12802.

Yang, Y. L., Loh, K. S., Liou, B. Y., Chu, I. H., Kuo, C. J., Chen, H. D., and Chen, C. S. (2013). SESN-1 is a positive regulator of lifespan in *Caenorhabditis elegans*. Exp Gerontol, 48(3), 371-379. doi:10.1016/j.exger.2012.12.011.